# Diet–Microbiota Interactions in Inflammatory Bowel Disease

**DOI:** 10.3390/nu13051533

**Published:** 2021-05-01

**Authors:** Kohei Sugihara, Nobuhiko Kamada

**Affiliations:** Division of Gastroenterology and Hepatology, Department of Internal Medicine, University of Michigan, Ann Arbor, MI 48109, USA; ksugihar@umich.edu

**Keywords:** inflammatory bowel disease (IBD), gut microbiota, diet–microbiota interactions, metabolic reprogramming, precision nutrition

## Abstract

Inflammatory bowel disease (IBD) is a chronic inflammatory disease of the gastrointestinal tract. Although the precise etiology of IBD is largely unknown, it is widely thought that diet contributes to the development of IBD. Diet shapes the composition of the gut microbiota, which plays critical roles in intestinal homeostasis. In contrast, intestinal inflammation induces gut dysbiosis and may affect the use of dietary nutrients by host cells and the gut microbiota. The interaction of diet and the gut microbiota is perturbed in patients with IBD. Herein, we review the current knowledge of diet and gut microbiota interaction in intestinal homeostasis. We also discuss alterations of diet and gut microbiota interaction that influence the outcome and the nutritional treatment of IBD. Understanding the complex relationships between diet and the gut microbiota provides crucial insight into the pathogenesis of IBD and advances the development of new therapeutic approaches.

## 1. Introduction

Inflammatory bowel disease (IBD), which includes ulcerative colitis (UC) and Crohn’s disease (CD), is a chronic and relapsing inflammatory disorder of the gastrointestinal tract [1]. The prevalence of IBD has been increasing worldwide, affecting about 3 million people in the United States and 2.5 million people in Europe [2,3]. Although the precise etiology of IBD has not yet been defined, it is widely accepted that the confluence of multiple factors, including genetic and environmental factors, is associated with its pathogenesis [1,4]. Genetic studies have identified over 200 host genetic loci associated with the risk of IBD, and mostly related to immunological pathways, including innate and adaptive immune responses and autophagy [1]. The prevalence of IBD is high in Western countries; however, the rates of IBD are also rising in many newly industrialized countries as they become more westernized. For example, the number of IBD patients has increased approximately 20-fold in the past 30 years in Japan [5]. This exponential increase suggests that environmental exposures also play critical roles in the development of IBD [6]. Among environmental factors, diet, smoking, stress, sleep patterns, hygiene, and antibiotic usage are considered to contribute to the risk of IBD [1]. In particular, diet is widely thought to have a pivotal role in the pathogenesis of IBD [7]. It is well-known that diet shapes the composition of the gut microbiota. Gut microbes use diet-derived nutrients for their growth and colonization in the gut. In contrast, host cells use microbial metabolites as energy sources and immunomodulatory agents to maintain intestinal homeostasis. This symbiotic relationship between the gut microbiota and the host is crucial for human health [8]. However, the intake of certain diets, such as the westernized diet characterized by high fat and low fiber, results in gut dysbiosis, thereby disrupting intestinal homeostasis and promoting inflammation of the gut [9]. Gut inflammation, in turn, influences the composition and function of the gut microbiota [10]. In this review, we discuss the complex reciprocal interactions between diet and the gut microbiota in the context of IBD. In particular, we focus on the metabolic reprogramming of host and microbial cells during inflammation and how these metabolic changes may optimize dietary interventions. We also highlight the future direction of microbiota-targeted dietary interventions to develop precision nutrition for the treatment of IBD.

## 2. The Role of Diet in the Pathogenesis of IBD

### 2.1. Epidemiological Study

Several epidemiological studies have examined the association between specific dietary factors and the risk of IBD. Early retrospective case-control studies identified a westernized diet characterized by a higher intake of meat and fats, particularly polyunsaturated fatty acids (PUFAs), and a lower intake of fiber, fruits, and vegetables as the risk for IBD pathogenesis [11]. However, several limitations, including recall and selection biases, are inherent in retrospective case-control studies. To address these limitations, several large prospective cohort studies, such as the European Prospective Investigation in Cancer and Nutrition (EPIC) study and the Nurses’ Health Study (NHS), have attempted to characterize the link between diet and the risk of IBD. The EPIC studies demonstrated that increasing the intake of docosahexaenoic acid (DHA) is associated with a lower risk of CD, whereas the dietary pattern of high consumption of sugar and soft drinks is related to a higher risk of UC [12,13]. The NHS showed that total fiber and zinc intake is inversely associated with the risk of CD but not UC [14,15]. Interestingly, the NHS identified the role of a gene–diet interaction in the development of IBD. In this context, the study demonstrated that the variants in *CYP4F3*, which are involved in PUFA metabolism, may modify the association between n-3 and n-6 PUFA intake and the risk of UC [16]. Specifically, high intake of n-3/n-6 PUFAs was associated with a reduced risk of UC in individuals with the GG/AG genotype at a SNP in *CYP4F3*, but not in those with the AA genotype [16]. This interaction between gene variant and diet may contribute to the inconsistent associations between dietary factors and the risk of IBD. Although further validation is required, these studies suggest that specific dietary factors are associated with the pathogenesis of IBD.

### 2.2. Animal Study

Epidemiological evidence is always biased by several factors; thus, animal models are used to test hypotheses that propose a relationship between dietary factors and intestinal inflammation. In the context of experimental model colitis, several colitis models, including chemical agent-induced colitis, spontaneous colitis by genetic modification, and adoptive T-cell transfer-induced colitis, are applied for animal research of IBD. These colitis models mimic some key immunological and histopathological features of IBD in humans. In chemical agent-induced colitis models, dextran sodium sulfate (DSS) is most frequently used to clarify the association between dietary factors and gut inflammation. Oral administration of DSS via drinking water induces severe colitis characterized by weight loss, bloody diarrhea, ulcer formation, loss of epithelial cells, resembling some features of flares in human UC [17]. Although the exact mechanism of DSS-induced colitis remains incompletely understood, it is believed that DSS exerts chemical damage to intestinal epithelial cells, which promotes bacteria translocation and activates host immune response [17]. Likewise, IL-10-deficient colitis model is frequently used mouse colitis model that mimics human CD [18]. IL-10 is an immunoregulatory cytokine that is essential for the maintenance of intestinal homeostasis, and mice deficient in IL-10 spontaneously develop severe inflammation in the cecum and the colon [18]. These models are valuable tools to assess the effect of dietary factors and we should choose the colitis model appropriately according to the purpose of experiments.

Animal studies have generally supported the observation that diet plays a crucial role in the etiology of intestinal inflammation. For example, a high-fat diet exacerbated intestinal inflammation in the IL-10-deficient colitis model [19]. In contrast, wild-type mice fed a high-fat diet exhibited only low-grade intestinal inflammation [20,21], suggesting that diet alone is insufficient to drive severe inflammation in the gut. Rather, genetic susceptibility augments the effect of dietary factors, leading to the development of severe intestinal inflammation.

Various animal models of colitis have identified several dietary components, including phytochemicals and food additives, that prevent or promote intestinal inflammation (Figure 1). Dietary nutrients and components have been shown to influence the gut microbiota, the mucosal barrier, and mucosal immunity, all of which modulate susceptibility to intestinal inflammation. For example, supplementation with certain amino acids, including tryptophan, arginine, glutamine, glycine, and histidine, attenuates intestinal inflammation by modulating mucosal immunity or the gut microbiota in rodent models’ colitis [22,23,24,25,26]. In contrast, a recent study demonstrated that the intake of high levels of dietary simple sugars, including glucose, fructose, or sucrose, exacerbates colitis in dextran sodium sulfate (DSS)-induced colitis mice and IL-10-deficient mice [27]. Dietary simple sugars rapidly alter the gut microbiota, particularly mucin-degrading bacteria, such as *Akkermansia muciniphila* and *Bacteroides fragilis*. Sugar-induced exacerbation of colitis is not observed in germ-free (GF) mice, implying that an altered gut microbiota plays a critical role.

However, animal study has some limitations, whereby the results obtained are not always translatable to the clinical setting. Certain types of diet are reported to be effective in both animal studies and human clinical studies. For example, the Crohn’s disease exclusion diet (CDED), which is designed to reduce exposure to dietary components that negatively affect the microbiome, ameliorates intestinal inflammation in animal models of IBD [28]. Consistently, the CDED effectively induces remission in patients with CD [28]. Further, the supplementation of certain dietary fibers as prebiotics induces remission in UC patients [29,30]. On the other hand, the therapeutic effect of certain dietary treatments observed in animal studies is not recapitulated in human clinical trials. For instance, some prebiotics that ameliorate symptoms in animal IBD models are ineffective in human IBD [31]. In clinical research, variations in host genetics, environmental factors, and the gut microbiota among study participants may impact the response to the dietary modification. Thus, personalized nutrition may be required to optimize the dietary interventions for patients with IBD.

## 3. Gut Microbiota and IBD

### 3.1. The Role of the Gut Microbiota in IBD

Accumulating evidence suggests that the gut microbiota plays pivotal roles in the regulation of intestinal homeostasis and IBD pathogenesis. GF mice have immunological defects in the gut, including fewer and smaller mesenteric lymph nodes and Peyer’s patches, and decreased numbers of T helper 17 (Th17) cells and regulatory T (Treg) cells in colonic lamina propria [32]. A defective colonic mucus barrier, which is a physiological barrier against luminal bacteria, is also evident in GF mice [33]. In contrast, mucosal immunity and mucus barrier functions can be restored by the colonization of the gut microbiota [33]. Thus, the gut microbiota is an essential factor for the proper development of intestinal immunity and intestinal barrier integrity. Despite this pivotal role in intestinal health, mounting evidence indicates that the gut microbiota drives intestinal inflammation in IBD [10]. Several genetically engineered animal IBD models, such as IL-10-deficient mice, spontaneously develop colitis, triggered by an excessive immune reaction against the commensal microbiota [18]. Notably, most IBD-prone animals never develop colitis if raised in GF conditions [34,35]. Likewise, the colonization by the dysbiotic gut microbiota isolated from IBD patients is sufficient to elicit abnormal immune activation and induce severe inflammation in the cecum and colon [36,37]. Conversely, the colonization by the gut microbiota from healthy subjects does not induce colitis [37], suggesting that IBD-associated dysbiosis contributes to the pathogenesis of IBD. To manipulate the gut dysbiosis in IBD, gut microbiota-targeting therapies, including probiotics, prebiotics, and fecal microbiota transplantation (FMT), have been studied. However, the efficacy of these treatment is limited [38,39], and thereby another therapeutic approach might be required to manipulate the gut dysbiosis to treat IBD.

### 3.2. Alteration of the Microbial Composition in IBD

It is well-known that gut dysbiosis is often observed in patients with IBD. Several studies have studied the differences in the gut microbial composition between IBD patients and healthy individuals. Genetic and environmental factors shape the gut microbiota, and intestinal inflammation also changes the microbial community in the gut. In microbiome studies, α- and β-diversity are used to estimate the microbial properties. Alpha diversity means the variation of microbiome within a sample, which include species richness and species diversity. In contrast, β-diversity is the variation of microbial community between the samples. In patients with IBD, reduced α-diversity, a decreased abundance of Firmicutes and Bacteroidetes, and an increase in Proteobacteria are the common features of gut dysbiosis [10]. Although the precise mechanisms by which gut dysbiosis develops in patients with IBD are incompletely understood, gut inflammation significantly changes the gut microenvironment, including nutrient availability and oxygen levels, which, in turn, contributes to the alteration of the microbial composition of the gut. For example, in the healthy gut, which is characterized by low oxygen levels, the dominant bacterial communities are obligate anaerobes, such as bacteria belonging to the phyla Firmicutes and Bacteroidetes [40]. However, gut inflammation increases oxygen levels in the gut lumen, thereby limiting the growth of obligate anaerobes [41]. In contrast, the microenvironmental alterations associated with intestinal inflammation promote the fitness of facultative anaerobes, including Proteobacteria [41]. Thus, inflammation-associated changes in luminal oxygen levels may trigger the gut dysbiosis associated with IBD.

Furthermore, it is widely accepted that certain pathogenic members of the gut microbiota, namely pathobionts, accumulate in the feces and mucosa of IBD patients. Although no single causative microorganism has been identified, many studies have reported the potential pathobionts involved in IBD pathogenesis. For instance, the accumulation of adherent and invasive *Escherichia coli* (AIEC), an IBD-associated pathobiont, in the ileal and colonic mucosa of IBD patients has been reported [42]. AIEC strains harbor several virulence genes related to adhesive and invasive properties in intestinal epithelial cells and can survive and replicate extensively in macrophages without triggering host cell death [42]. The colonization by AIEC causes mild inflammation in healthy mice [43], and massive intestinal inflammation in genetically susceptible mice and chemical-induced colitis mice [44,45]. Moreover, persistent AIEC colonization promotes intestinal fibrosis by way of flagellin-mediated activation of IL-33–ST2 signaling in *Salmonella*- or DSS-induced colitis mice [46], suggesting a close association between AIEC and the augmented inflammation and fibrosis in the context of IBD.

Recent studies have revealed that oral gut bacteria are associated with the pathogenesis of IBD [47,48]. It has been reported that oral resident bacteria, such as Fusobacteriaceae and Pasteurellaceae, are accumulated in the intestinal mucosa of patients with IBD [49,50]. Likewise, the colonization by *Klebsiella* species, isolated from the saliva of CD patients, into GF mice strongly induces Th1 cells, which in turn, elicit severe gut inflammation [51], indicating that certain oral resident bacteria act as pathobionts. In the context of the oral–gut axis, the prevalence of periodontitis is significantly higher in IBD patients compared with non-IBD patients [52]. Consistent with this notion, periodontal inflammation markedly exacerbates gut inflammation in mice by facilitating ectopic gut colonization by oral pathobionts [53]. Further, in the presence of periodontal inflammation, oral pathobiont-reactive Th17 cells arise in the oral mucosa and migrate to the inflamed gut, promoting gut inflammation in experimental colitis mice [53]. Thus, besides gut dysbiosis, the perturbation of the oral microbiota may likewise contribute to the pathogenesis of IBD. The mechanism by which IBD-associated pathobionts promote gut inflammation has been elucidated; however, therapeutic approach for suppressing pathobiont remains poorly studied. Thus, future study should clarify the therapeutic approach targeting IBD-associated pathobionts.

### 3.3. Functional Changes in the Gut Microbiota in IBD

Advances in multiomics technologies, including metagenomics, metatranscriptomics, and metabolomics, explore the disease landscape by investigating different aspects of the functions of the gut microbiota. As multiomics provide a global view of changes in the metabolic and biochemical capacities of the gut microbiome, the focus of current research is shifting towards the functional alteration in disease conditions.

Although the taxonomic composition of the gut microbiota is dramatically varied among individuals, metagenomic analysis has revealed a more conserved functional composition [54]. In the gut microbiota of IBD, metagenomic studies have identified significant shifts in oxidative stress pathways and reduced carbohydrate metabolism and amino acid biosynthesis [55,56]. However, metagenomics is limited to revealing the functional potential, not the functional activity, of microorganisms. In this context, metatranscriptomics enables one to measure actual gene expression and evaluate the functional activity of the gut microbiota. Microbial transcriptions rapidly respond to environmental stimulation, such as host immune activation and inflammation. For example, acute innate and adaptive immune stimulation achieved by treatment with flagellin or anti-CD3 antibody induce dramatic transcriptional reprogramming of the commensal bacteria [57]. Acute immune responses upregulate the genes involved in stress responses and downregulate the genes associated with carbohydrate utilization and amino acid biosynthesis in commensal bacteria [57]. Microbial transcriptions induced by immune activation are accompanied by an altered production of bacterial metabolites, such as short-chain fatty acids (SCFAs) and amino acids [57]. Likewise, intestinal inflammation markedly changes transcriptional profiles, including stress response and nutrient metabolism, in IBD-associated AIEC [58]. Importantly, these stimulations do not alter the relative abundance of bacteria; rather, the gut bacteria rapidly change gene transcription to adapt to the particular environment.

In human studies, metatranscriptional profiles are more varied between individuals than metagenomic functional profiles [59], suggesting that measuring actual gene expression is vital to evaluate the functional activity of the gut microbiota. A study that applied metagenomic and metatranscriptomic profiling of the gut microbiomes of more than 100 individuals found that the abundances of many bacterial functional genes correlated well at both DNA and RNA levels [60]. However, some bacterial genes abundant in the metagenomic analysis were transcriptionally inactive when analyzed at the RNA level [60], suggesting that taxonomic abundance does not always correlate with metabolic activity. Additionally, a recent multiomics study showed a comprehensive view of functional dysbiosis, such as reduced SCFA and vitamin production and decreased deconjugation of bile acids, in the gut microbiota of IBD patients [61] (Figure 2). These functional changes in the gut microbiota are associated with the efficacy of treatment. For example, metabolic interactions, especially SCFA synthesis, were reduced in the gut microbiota from patients who did not respond to anti-TNF therapy compared to the gut microbiota from responders [62]. These studies suggest that microbial metabolic activities are critical for disease pathogenesis and therapeutic efficacy.

## 4. Diet–Microbe Interaction in IBD

Gut bacteria mainly use diet-derived nutrients and produce various microbial metabolites that exert important and diverse effects on intestinal homeostasis (Figure 3). The importance of the interaction between diet and the gut microbiota in intestinal homeostasis has been illustrated by GF mice, as described in Section 2.2. In patients with IBD, gut dysbiosis is accompanied by disruption of diet–microbe interactions that may influence the efficacy of dietary intervention. In this section, we discuss the current knowledge of diet–microbe interactions in intestinal homeostasis and gut inflammation.

### 4.1. Dietary Fibers and SCFAs

SCFAs, including acetate, propionate, and butyrate, are the most well-studied microbial metabolites that display a wide variety of biological functions in the gut. SCFAs are mainly derived from bacteria-accessible dietary fibers. Gut bacteria-produced SCFAs are key energy substrates for colonocytes, and in general, SCFAs act as signaling molecules by way of G protein-coupled receptors (GPRs) in various types of host cells [63]. In addition, SCFAs, particularly butyrate, are well-known to regulate gene expression epigenetically by inhibiting histone deacetylases (HDACs), which control the expression of numerous genes [64]. For example, butyrate enhances histone H3 acetylation by inhibiting HDACs in naïve CD4 T cells, which in turn, upregulates the expression of Foxp3 and promotes Treg cell differentiation [65]. Moreover, butyrate promotes anti-inflammatory properties in macrophages and dendritic cells by activating GPR109a, which enables these cells to support the differentiation of Treg cells [66]. Indeed, butyrate-mediated mucosal immunity improves colitis in the murine T cell transfer model of colitis [65]. Similarly, acetate, another SCFA, also regulates the inflammatory response via GPR43, suggesting that this particular SCFA is critical to regulate intestinal immunity [67].

In addition to their immune regulatory function, SCFAs are also important for mucosal barrier function. Butyrate regulates the integrity of epithelial tight junctions and the secretion of mucus, thus enhancing mucosal barrier function [68,69]. In contrast, dietary fiber regulates the function of the mucosal barrier independently of SCFAs. A low-fiber diet decreases microbial diversity and SCFA production, but also shifts the gut microbial metabolism toward the utilization of less favorable substrates, particularly host-secreted mucins [70]. A lack of dietary fibers upregulates mucin degradation activity, as well as the expansion of mucin-foraging bacteria, such as *A. muciniphila*, which impair mucosal barrier function [70]. Consistent with this, the Western diet characterized by high fat and low fiber increases the penetrability of the inner mucus layer in mice, and thus increases the susceptibility to infections [71].

As mentioned in Section 3.3, IBD-associated gut dysbiosis is accompanied by the reduced abundance of butyrate-producing bacteria, such as *Roseburia hominis* and *Faecalibacterium prausnitzii*, and, therefore, fecal SCFA levels are lower in IBD patients compared to healthy individuals [61]. In addition to microbial function, the utilization of SCFAs by host cells is also changed in patients with IBD. For example, proinflammatory cytokines TNF and IFN-γ decrease the expression of monocarboxylate transporter MCT1, a butyrate transporter, and further, MCT1 expression is markedly decreased in the inflamed colonic mucosa of IBD patients [72]. Several studies have demonstrated that butyrate oxidation and the expression of genes involved in butyrate oxidation are diminished in the intestinal mucosa of IBD patients [73,74,75]. These studies suggest that the utilization of SCFAs by host cells and gut bacteria are diminished in IBD, particularly in the inflamed mucosa. Although several animal studies have reported that the supplementation of dietary fiber attenuates intestinal inflammation [76,77], there is a little clinical evidence to support the efficacy of this practice in patients with IBD. The supplementation of certain fibers alleviates disease activity in UC [29,30], whereas prebiotic fructo-oligosaccharide is apparently ineffective in CD [31]. In addition, small clinical trials have questioned the efficacy of SCFA enemas for treating UC [78,79]. The efficacy of fiber and SCFA supplementation may be related to the metabolic activity of the gut microbiota or host cells; hence, further studies are necessary to identify the key factors of any successful therapeutic efficacy for IBD.

### 4.2. Dietary Fat and Bile Acids

Bile acids are synthesized from cholesterol in hepatocytes and secreted into the duodenum. Although most bile acids are transported back into the liver by the enterohepatic circulation, about 5% of bile acids escape reabsorption in the ileum and are subject to bacterial transformation in the colon. Bile acids have been known to facilitate the absorption of lipids or fat-soluble vitamins in the small intestine. Moreover, bile acids regulate host energy metabolism by acting on a family of cell membrane and nuclear receptors, such as the farnesoid X receptor (FXR) and the vitamin D receptor (VDR) [80].

In addition to modulating host metabolism, recent studies have reported that bile acid metabolites play a critical role in adaptive immunity, particularly Treg cell differentiation, which influences susceptibility to colitis. Distinct metabolites of bile acids are implicated in regulating specific T cell responses. For example, 3-oxo lithocholic acid (LCA) inhibits the differentiation of Th17 cells by directly binding to retinoid-related orphan receptor γt (RORγt), whereas isoalloLCA promotes the differentiation of Treg cells through the production of mitochondrial reactive oxygen species [81]. Furthermore, secondary bile acid 3β-hydroxydeoxycholic (IsoDCA) acid also promotes the generation of Treg cells by suppressing the immunostimulatory properties in dendritic cells [82]. In the context of diet, dietary components influence the profile of bile acids and T cell responses. Compared with grain-based standard chow, a purified diet (use refined ingredients) decreases RORγ^+^ Treg cells and certain secondary bile acids [83]. In contrast, treatment with a mixture of certain primary or secondary bile acids can restore colonic RORγ^+^ Treg cells through bile acid–VDR signaling. Interestingly, the induction of RORγ^+^ Treg cells by bile acid metabolites is restricted to the colon [83], suggesting that the interaction of bile acids and T cells is critical in regulating colonic inflammation. In fact, supplementation of the primary or secondary bile acid mixture in drinking water ameliorates gut inflammation in a VDR-dependent manner [83].

It has been well-documented that dietary fat content influences the profile of bile acids. A milk-derived diet high in saturated fats promotes hepatic taurine conjugation of bile acids in IL-10-deficient mice, which leads to the accumulation of sulphite-reducing bacteria, including *Bilophila wadsworthia* [84]. A high-fat diet-induced expansion of *B. wadsworthia* exacerbates Th1-mediated inflammation in IL-10-deficient mice [84]. Although a high-fat diet is known to reduce Treg cells in the colon [19], the role of the interaction between bile acids and Treg cells in response to dietary fat in gut inflammation remains unknown. Dietary fat is most interested nutrients for the treatment of IBD; however, there are inconsistent evidence regarding the impact of dietary fat in IBD. For example, there are no significant difference between elemental diet and polymeric diet that contain different amount of fat [85]. In contrast, a recent study reported that a low-fat diet improves the marker of inflammation and gut dysbiosis in patients with UC [86]. Further study needs to clarify the impact of dietary fat in IBD as well as the interaction between bile acid metabolites and host immunity.

### 4.3. Dietary Tryptophan and Indole Derivatives

Tryptophan is an essential amino acid and a common constituent of protein-rich foods, such as fish, meat, and cheese. Tryptophan is a precursor for some bioactive metabolites, such as kynurenine and serotonin. Kynurenine, an endogenous tryptophan metabolite, is well-known to act as a ligand for the aryl hydrocarbon receptor (AhR), a critical regulator of immunity and inflammation involved in adaptive immunity and mucosal barrier function [87]. In addition, tryptophan metabolites generated by the gut microbes also contribute to the regulation of mucosal homeostasis. Tryptophan can be metabolized by certain gut bacteria, such as lactobacillus, into a range of indole metabolites, some of which can act as AhR ligands [88]. Indole metabolites play a role in mucosal immune responses via AhRs by modulating the production of IL-22, a cytokine with well-known effects on intestinal homeostasis [88,89]. In fact, supplementation with dietary tryptophan and indole metabolites protects against colitis through AhR [22].

A genome-wide association study found that caspase recruitment domain-containing protein 9 (CARD9), an adaptor protein involved in apoptosis and antifungal immunity, is encoded by a susceptibility gene for IBD [90]. *Card9*-deficient mice exhibit reduced expression of IL-22 in the colon, and are more susceptible to colitis [91]. Interestingly, certain bacteria, such as *Lactobacillus reuteri* and *Allobaculum*, that are capable of catabolizing tryptophan into indole derivatives, are decreased in *Card9*-deficient mice. Gut dysbiosis in *Card9*-deficient mice decreases AhR activation and hinders recovery from colitis. Consistent with animal experiments, AhR activity and tryptophan metabolites are reduced in IBD patients, particularly in those with the *CARD9* risk alleles associated with IBD [91]. Thus, IBD-associated gut dysbiosis decreases tryptophan metabolism, which in turn, affects IL-22-mediated mucosal immunity.

### 4.4. Dietary L-Serine

L-serine is a nonessential amino acid that can be synthesized in the liver. L-serine plays a critical role in several metabolic processes in mammalian cells, especially in disease conditions [92]. For example, L-serine metabolism is markedly upregulated in cancer cells and activated immune cells, including T cells and macrophages, as cells require L-serine to proliferate and survive [93,94,95]. Likewise, gut bacteria also use L-serine in certain conditions. In the inflamed gut, AIEC upregulates the transcription of L-serine transport and metabolism genes, which conveys a growth advantage over commensal *E. coli* [58]. Interestingly, the concentration of luminal L-serine is largely dependent on dietary intake; thus, depriving a diet of L-serine can suppress inflammation-induced blooms of AIEC [58], suggesting that AIEC uses dietary L-serine for growth and fitness in the inflamed gut. Hence, dietary L-serine deprivation attenuates intestinal inflammation by suppressing the bloom of Enterobacteriaceae in IL-10-deficient mice colonized by gut microbiota from CD patients [58]. In contrast, the rectal supplementation of L-serine also attenuates intestinal inflammation in DSS-induced colitis mice [96]. These controversial results imply that the gut microbiota, in particular the presence or absence of bacteria that use L-serine, has an impact on the effect of dietary L-serine in gut inflammation.

### 4.5. Dietary Emulsifiers

The widespread use of processed food has increased the consumption of food additives, such as emulsifiers and artificial sweeteners. Food additives are generally believed to be safe; however, recent studies have shown that some food additives influence the microbial community and gut inflammation [97]. Emulsifiers, which are the most researched food additives, are incorporated into processed foods to enhance texture and stability [98]. Studies have shown that emulsifiers, including carboxymethylcellulose and polysorbate-80 disrupt host–microbe interactions, resulting in a microbiota with enhanced mucolytic and proinflammatory activity that promotes mucus degradation and intestinal inflammation [99]. In contrast, these particular effects of emulsifiers are not observed in GF mice nor in altered Schaedler flora (ASF) mice [99,100], both of which are colonized by low-complexity microbiota, suggesting that certain bacteria are necessary for emulsifier-induced gut inflammation. Interestingly, emulsifiers directly alter the human gut microbiota, increasing proinflammatory potential in the gut microbiota [100]. Investigators have shown that emulsifiers directly change the expression of virulence genes related to bacterial motility and adhesion in AIEC [101]. The colonization by AIEC in ASF mice was sufficient to promote the detrimental effects of emulsifiers. Notably, a small clinical trial that evaluated the feasibility and acceptability of a low emulsifier diet provided no clinical evidence of the effectiveness of dietary emulsifiers on disease activity in IBD [102]. Therefore, further study is needed to validate the effect of dietary emulsifiers on the gut microbiota and IBD in humans.

## 5. The Impact of Nutritional Intervention on the Gut Microbiota in IBD

Multiple dietary components have been shown to suppress or aggravate inflammation in IBD. Thus, dietary manipulation as an adjunctive or replacement treatment strategy for IBD has been highly pursued. To date, several nutritional interventions, including enteral nutrition and diet modification, are being used for the treatment of IBD [103]. These nutritional interventions have been shown to promote clinical remission and mucosal healing. Despite a limited understanding of the precise mechanisms, nutritional interventions clearly impact the gut microbiota (Table 1), and thus, manipulation of gut microbiota is thought to be one of the mechanisms.

### 5.1. Enteral Nutrition

Exclusive enteral nutrition (EEN) is the most extensively researched nutritional intervention known to induce remission in CD patients. A recent meta-analysis showed that EEN is more effective in inducing remission than corticosteroids in pediatric patients with CD [85]. Moreover, an open-label randomized controlled trial (RCT) demonstrated that EEN more successfully promotes mucosal healing compared to corticosteroids in pediatric CD [110]. In contrast, enteral nutrition is less effective than corticosteroids in adult patients with CD due to low palatability [85]. The efficacy of EEN has been attributed to several mechanisms, including bowel rest, anti-inflammatory effects, alteration of the gut microbiota, and recovery of the intestinal epithelial barrier [111]. Profound changes in the composition of the gut microbiota induced by EEN have been reported [104,105,106]. Although EEN seems to correct IBD-associated gut dysbiosis, the microbial community resulting from EEN treatment differs markedly from that in healthy individuals. Consistent with this notion, most studies have shown that EEN treatment reduces α-diversity and beneficial bacteria, such as *Faecalibacterium* and *Bifidobacterium* [104,105]. However, the precise mechanisms by which alteration of the fecal microbial community during EEN attenuates intestinal inflammation remains unclear. Further study of the relationship between the alteration of a microbial community and the efficacy of EEN is required. A better understanding of the mechanisms involved could lead to new therapeutic strategies for a dietary approach to IBD.

### 5.2. Dietary Intervention

New dietary approaches that improve the palatability associated with the efficacy of EEN treatment have been developed in recent years. The Crohn’s disease exclusion diet (CDED), which is a whole-food diet, coupled with partial enteral nutrition (PEN), was designed to reduce exposure to dietary components that are hypothesized to negatively influence inflammation in IBD [112]. It has been shown that CDED plus PEN induces remission in both children and adults with CD, including in patients with secondary loss of response to anti-TNF therapies [113]. A multinational RCT demonstrated a comparable remission rate between CDED plus PEN and EEN in pediatric patients with CD, and higher tolerance of CDED plus PEN than EEN [28]. In addition, the sustained remission rate was higher with CDED plus PEN than with a free diet plus PEN. Both CDED plus PEN and EEN changed microbial communities in patients who achieved remission, whereas patients who did not achieve remission exhibited a lesser change in the microbial community [28], suggesting that microbial manipulation is associated with the improvement of inflammation.

To mimic the effect of EEN on the gut microbiota, a CD treatment-with-eating diet (CD-TREAT) was devised [109]. CD-TREAT and EEN similarly affect the gut microbiota and microbial metabolites in healthy subjects [109]. Furthermore, a small clinical trial in pediatric patients with active CD showed that CD-TREAT induces a clinical response in 80% (4/5) and clinical remission in 60% (3/5) with a significant concurrent decrease in fecal calprotectin [109]. Some effects induced by EEN and CD-TREAT may be linked to the alteration of the gut microbiota. However, the clinical responses do not always correlate with the changes in beneficial bacteria and bacterial metabolites, illustrating the puzzling nature of the mechanisms of action of these dietary treatments. For example, EEN and CD-TREAT each decrease fecal SCFA levels in healthy individuals [104,109], even though SCFAs are known beneficial metabolites generated by the gut microbiota. Thus, the paradox between the clinical response and the alteration of gut microbiota remains a conundrum in the context of the mechanistic understanding of dietary interventions, including EEN therapy.

Notably, other dietary interventions, including the low fermentable, oligo-, di-, monosaccharides and polyols (FODMAP) diet, a specific carbohydrate diet, and a low-fat diet, also impact the gut microbiota, disease activity, and gastrointestinal symptoms in patients with IBD [86,107,114]. Nevertheless, a recent meta-analysis has shown that the effects of dietary interventions on IBD are uncertain due to the lack of high-quality clinical trials [115]. Consequently, there is an urgent need for RCTs to evaluate the efficacy of dietary interventions and their effect on the microbiota.

## 6. Conclusions and Future Direction

Multiple recent studies have highlighted the roles of complex crosstalk between diet and the gut microbiota in intestinal homeostasis. Intestinal inflammation changes the microenvironment, altering the composition and function of the bacterial community in the gut. Clearly, the diet–microbe interaction is more complicated in IBD [116]. A disturbance of the interaction between diet and the gut microbiota may be associated with the efficacy of treatment for IBD. The function of the gut microbiota is heterogeneous in IBD patients, requiring personalized nutrition approaches adapted to the characteristics of genetic factors, clinical background, and the gut microbiota [117]. This paradigm of precision nutrition is an emerging concept of the next-generation nutrition therapy for IBD. Understanding the complex network of interactions between diet and microbiota will advance the field of precision nutrition. In addition, the development of predictive tools and biomarkers will help identify subgroups of IBD patients who are responders and nonresponders to nutritional intervention.

## Figures and Tables

**Figure 1 nutrients-13-01533-f001:**
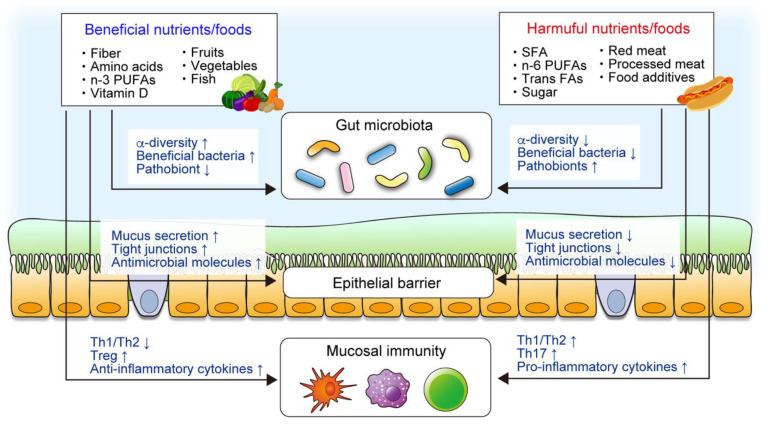
The roles of nutrients and foods in the pathogenesis of inflammatory bowel disease (IBD). Epidemiological, clinical, and animal studies have demonstrated that certain components of diet are associated with IBD. Diet plays a critical role in intestinal homeostasis, including the gut microbiota, intestinal mucosal barrier, and mucosal immune system. Diet directly modulates the mucosal barrier and immunity, whereas diet–microbiota interaction also regulates the intestinal homeostasis.

**Figure 2 nutrients-13-01533-f002:**
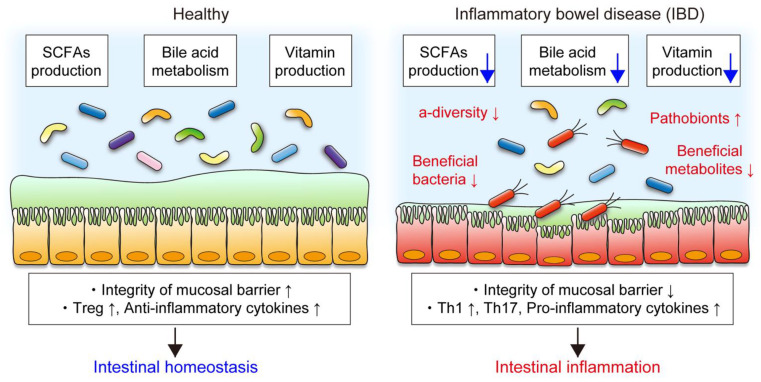
Functional changes in the gut microbiota in IBD. The gut microbiota produces short-chain fatty acids (SCFAs) and vitamins and deconjugates bile acids, which play crucial roles in mucosal barrier and immunity. In patients with IBD, decreased α-diversity, reduced abundance of beneficial bacteria and metabolites, and increased pathobionts are the common features of gut dysbiosis. Gut dysbiosis is accompanied by functional alteration of the gut microbiota. In particular, decreased SCFA production, bile acid metabolism, and vitamin production in the gut microbiota in IBD are associated with impaired mucosal barrier integrity and abnormal immune reactions, resulting in intestinal inflammation.

**Figure 3 nutrients-13-01533-f003:**
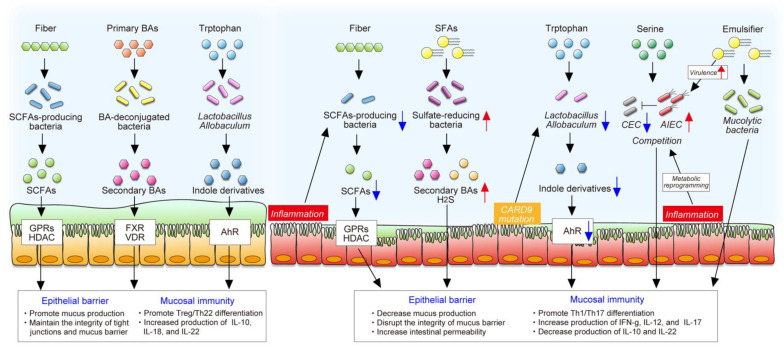
The roles of diet–microbe interaction in intestinal homeostasis and inflammation. The gut microbiota produces various metabolites from dietary components that exert important and diverse effects on intestinal homeostasis, including the maintenance of the epithelial barrier and mucosal immunity. In contrast, gut inflammation or mutation of certain genes change microbial composition and function, decreasing the production of beneficial microbial metabolites, such as SCFAs and indole derivatives. In addition, inflammation-induced metabolic reprogramming of adherent and invasive *Escherichia coli* (AIEC) enables it to adapt to the inflamed gut and to compete with commensal *E. coli* (CEC). Westernized diet, characterized by high saturated fat and emulsifiers, promotes blooms of certain pathobionts, some of which impair the epithelial barrier and stimulate a proinflammatory response.

**Table 1 nutrients-13-01533-t001:** The impact of dietary intervention on the gut microbiota in IBD.

Year	Study Design	Subject	Nutritonal Intervention	Results		Ref
				Clinical	Gut Microbiota/Metabolites	
2014	Cohort	CD: 15, HC: 21	EEN (PD)	no data	Diversity ↓, *Faecalibacterium prausnitzii* ↓, butyrate ↓, sulfide ↑	[104]
2015	Cohort	CD: 23, HC: 21	EEN (PD)	62% remission rate	Diversity ↓, *Bifidobacterium* ↓, *Ruminococcus* ↓, *Faecalibacterium* ↓, *Akkermansia* ↓, *Lactococcus* ↑	[105]
2015	Cohort	CD: 90	EEN vs. PEN vs. Anti-TNF	45% clinical response	*Haemophilus* ↓, *Streptococcus* ↓, *Dialister* ↓, *Dorea* ↓, *Gordonibacter* ↓, *Alistipes* ↑	[106]
2020	RCT	IBD: 52	Low-FODMAP diet vs. control diet	Relief of GI symptoms ↑, IBS score ↓, HR-QOL ↑	*Bifidobacterium adolescentis* *↓, B. longum* *↓, Faecalibacterium prausnitzii* *↓, B. dentium* *↑*	[107]
2016	Cross-over	CD: 9	Low-FODMAP diet vs. Australian diet	GI symptoms ↓, Fecal calprotectin →	*Clostridium* cluster XIVa ↓, *Akkermansia muciniphila* ↓, *Ruminococcus torques* ↑, SCFA →	[108]
2019	RCT	CD: 78	CDED + PEN vs. EEN (PD)	% remission rate→, sustained remission ↑, tolerability ↑	*Haemophilus* ↓, *Veillonella* ↓, *Bifidobacterium* ↓, *Prevotella* ↓, *Anaerostipes* ↓, *Oscillibacter* ↑, *Roseburia* ↑	[28]
2020	Cross-over	UC: 17	Low-fat diet vs. improved standard American diet	QOL ↑, amyloid A ↓	Actinobacteria ↓, Bacteroidetes ↑, *Faecalibacterium prausnitzii* ↑, *Prevotella* ↑, acetate ↑, Trp ↑, lauric acid ↓	[86]
2019	RCT/cohort	HC: 25 (microbiota) CD: 5 (clinical)	CD-TREAT vs. EEN (PD) vs. habitual diet	80% clinical response, fecal calprotectin ↓	*Prevotella* ↑, *Escherichia Shigella* ↑, *Eisenbergiella* ↑, *Lachnoclostridium* ↑, *Bifidobacterium* ↓, *Faecalibacterium* ↓, *Ruminococcus* ↓	[109]

RCT, randomized controlled trial; IBD, inflammatory bowel disease; CD, Crohn’s disease; HC, healthy control; EEN, exclusive enteral nutrition; PEN, partial enteral nutrition; PD, polymeric diet; FODMAP, fermentable oligo-, di-, monosaccharides and polyols; CDED; Crohn’s disease exclusion diet; CD-TREAT, CD treatment-with-eating diet; GI symptoms, gastrointestinal symptoms; IBS, irritable bowel syndrome; HR-QOL, health related-quality of life; SCFA, short chain fatty acidsTrp, tryptophan; ↓, decrease; ↑, increase.

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
