# Peer review of "Diet–Microbiota Interactions in Inflammatory Bowel Disease"

_nutrients, 2021, doi:10.3390/nu13051533_

Round 1

Reviewer 1 Report

This review is a timely and comprehensive summary of current knowledge on the effect of diet and dietary components in the development and pathogenesis of IBD in the context of the microbiota. While in some areas, the review is a little superficial, the overall scope is quite broad and so this is to be expected.

Strengths:

  1. The writing style and quality is excellent
  2. The three figures are very helpful and well made
  3. The structure and organization is good
  4. The review will be an excellent resource for those beginning to study the effect of microbiota in IBD

Suggestions for improvement:

  1. In the introduction, it is twice mentioned that IBD is increasing worldwide. It would be good to include some statistics on incidence etc in places other than the US
  2. Section 2.1 explains details of studies but lacks a conclusion/summary of the concept as a whole – this is true of other sections as well – this is the opportunity for the author to put their own perspective on findings, which increases the value of a review
  3. Section 2.2 – additional text briefly describing the relevant mouse models would be useful – how they are induced and how well they reflect human disease
  4. Section 3.1 “defects in Tregs” – please describe what these defects are
  5. Line 164, and others – in many places it is difficult to understand what the experimental design was in referenced studies, in particular, whether they are mouse or human based, or in this case (ref 45) human bacteria INTO mice. This makes it difficult to interpret some of the discussed points. A review of the whole document to clarify these details is necessary
  6. Line 278 – use TNF not TNF-a (doi.10.1001/jamadermatol.2015.4322)
  7. Line 302 – “bile acids control adaptive immune responses” – this is pretty bold and I suspect not what was intended. Please rephrase to make the same point but in context of the disease/processes
  8. Line 310 – what is meant by a “purified” diet?
  9. Some discussion of measures of bacterial diversity (species frequency, alpha and beta diversity etc) somewhere in the manuscript would improve the discussion.

Author Response

Point-by-Point Responses to Reviewers
We are grateful for the thorough review and highly constructive comments. To address the
points raised by the reviewers, we have revised the main text and figures accordingly.
Reviewer 1:
Comment: In the introduction, it is twice mentioned that IBD is increasing worldwide. It would be
good to include some statistics on incidence etc in places other than the US
Reply:
We have added the information about the number of patients in Europe and Japan (Page 2, line
25 and page 2, line 32-34)
Comment: Section 2.1 explains details of studies but lacks a conclusion/summary of the concept
as a whole – this is true of other sections as well – this is the opportunity for the author to put their
own perspective on findings, which increases the value of a review
Reply:
We have added the conclusion or summary in the last of each section (Page 3, line 72-73, page
5, line 154-158, page 6, line 207-210, page 10, line 358-364)
Comment: Section 2.2 – additional text briefly describing the relevant mouse models would be
useful – how they are induced and how well they reflect human disease
Reply:
We have added the information regarding animal model colitis (Page 3, line 77-93)
Comment: Section 3.1 “defects in Tregs” – please describe what these defects are
Reply:
We have revised this sentence as follows: GF mice have immunological defects in the gut,
including fewer and smaller mesenteric lymph nodes and Peyer's patches, and decreased
numbers of T helper 17 (Th17) cells and regulatory T (Treg) cells in colonic lamina propria. (Page
5, line 138-141)
Comment: Line 164, and others – in many places it is difficult to understand what the experimental
design was in referenced studies, in particular, whether they are mouse or human based, or in this
case (ref 45) human bacteria INTO mice. This makes it difficult to interpret some of the discussed
points. A review of the whole document to clarify these details is necessary
Reply:
We have provided the experimental design to improve the clarity.
Comment: Line 278 – use TNF not TNF-a (doi.10.1001/jamadermatol.2015.4322)
Reply:
We have replaced “TNF-a” with “TNF” in the manuscript.
Comment: Line 302 – “bile acids control adaptive immune responses” – this is pretty bold and I
suspect not what was intended. Please rephrase to make the same point but in context of the
disease/processes
Reply:
We have revised this sentence as follows: In addition to modulating host metabolism, recent
studies have reported that bile acid metabolites play a critical role in adaptive immunity, particularly
Treg cell differentiation, which influences susceptibility to colitis. (Page 9, line 333-335)
Comment: Line 310 – what is meant by a “purified” diet?
Reply: Purified diet is a diet consists of refined ingredients. We have added the information about
the diet in this sentence. (Page 9, line 342-344)
Comment: Some discussion of measures of bacterial diversity (species frequency, alpha and beta
diversity etc) somewhere in the manuscript would improve the discussion.
Reply: We have added the discussion about microbial diversity in section 3.2. (Page 5, line 163-
167)

Reviewer 2 Report

A complete and informative review about the interaction between different diets and microbiota in the context of inflammatory bowel diseases; 

I have some queries:

A materials and methods section should be added to the study, describing what keywords were used in the research and what databases (Pubmed, Scopus-Embase, etc...) were used for this review.

Page 1 line 22-24 "Inflammatory bowel disease (IBD), which includes ulcerative colitis (UC) and Crohn’s disease (CD), is a chronic and relapsing inflammatory disorder of the gastrointestinal tract." you should add a citation, such as: doi: 10.1111/dth.12811. 

Page 1 line 30-32 "The prevalence of IBD is high in Western countries; however, the rates of IBD are also rising in many newly industrialized countries as they become more westernized, suggesting that environmental exposures also play critical roles in the development of IBD. " you should add a citation, such as: doi: 10.1371/journal.pone.0241575.

Thank You

Author Response

Point-by-Point Responses to Reviewers
We are grateful for the thorough review and highly constructive comments. To address the
points raised by the reviewers, we have revised the main text and figures accordingly.
Reviewer 2:
Comment: A materials and methods section should be added to the study, describing what
keywords were used in the research and what databases (Pubmed, Scopus-Embase, etc...) were
used for this review.
Reply:
This review is not a meta-analysis or systematic review. Other review articles in the Nutrients do
not have the material and method section. Thus, we think that we do not need to provide
information about keywords and databases.
Comment: Page 1 line 22-24 "Inflammatory bowel disease (IBD), which includes ulcerative colitis
(UC) and Crohn’s disease (CD), is a chronic and relapsing inflammatory disorder of the
gastrointestinal tract." you should add a citation, such as: doi: 10.1111/dth.12811.
Reply:
We have added a relevant citation in the sentence (Page 2, line 22-24).
Comment: Page 1 line 30-32 "The prevalence of IBD is high in Western countries; however, the
rates of IBD are also rising in many newly industrialized countries as they become more
westernized, suggesting that environmental exposures also play critical roles in the development
of IBD. " you should add a citation, such as: doi: 10.1371/journal.pone.0241575.
Reply:
We have added a relevant citation in the sentence (Page 2, line 33-35).

Round 2

Reviewer 2 Report

The authors responded to all queries. The paper is publishable